# Plant Melatonin: Regulatory and Protective Role

**Runxian Song [1,2], Faujiah Nurhasanah Ritonga [3], Haiyang Yu [2], Changjun Ding [4,*] and Xiyang Zhao [1,2,*]**

1   Jilin Provincial Key Laboratory of Tree and Grass Genetics and Breeding, College of Forestry and Grassland, Jilin Agricultural University, Xincheng Street No. 2888, Changchun 130118,

2   State Key Laboratory of Tree Genetics and Breeding, Forestry College, Northeast Forestry University, Harbin 150040, China

3   Institute of Vegetables, Shandong Academy of Agricultural Sciences, Jinan 250100, China

4   State Key Laboratory of Tree Genetics and Breeding, Key Laboratory of Tree Breeding and Cultivation of State Forestry Administration, Research Institute of Forestry, Chinese Academy of Forestry, Beijing 100091, China

*   Correspondence: changjund@126.com (C.D.); zhaoxyphd@163.com (X.Z.)

**Abstract:** **Melatonin** is an antioxidant that is widely distributed in plants and animals. It is a conservative molecule. In early studies, scientists often used isolation and identification techniques to observe whether the endogenous melatonin cycle in plants was related to the external photoperiod, plant growth, and development cycles, including seed germination, plant rooting, and floral induction. With the development of isolation and identification technology, there is ample evidence that plants possess a variety of melatonin-synthesis pathways. The comprehensive application of molecular biology, genomics, and computational biology has also led to a comprehensive understanding of the physiological functions of plant melatonin. In this paper, we not only highlight the candidate genes from *Arabidopsis thaliana* and *Oryza sativa* that might be contributing to increasing plant endogenous melatonin but also elucidate and characterize the role of melatonin in plant growth and development in response to biotic and abiotic stresses.

**Keywords:** abiotic stress; biosynthetic; biotic stress; melatonin; function model

## 1. Introduction

In 1917, a substance was extracted from the pineal gland of cattle (a small endocrine gland in the epithalamus of vertebrates) that could be used to cause the melanin in the skin of frogs to fade and turn white [1]. A few decades later, Lerner succeeded in identifying the chemical structure of the substance, naming it melatonin. Melatonin, known as N-acetyl-5-methoxytryptamine, is an indole compound with molecular formula $C_{13}H_{16}N_2O_2$ and a molecular weight of 232.27. The substance has amphiphilicity and is a highly lipophilic and partially hydrophilic compound.

Although melatonin has long been thought to be associated with vertebrates, since the 1990s, it has also been detected in invertebrates, higher plants, bacteria, and fungi [2], indicating that melatonin is a class of conserved small molecules that are ubiquitous in animals, plants, and microorganisms [3]. At present, scientists have found melatonin in most short-day plants [4].

In previous research, scientists determined the function of melatonin by identifying cyclical changes in the amount of endogenous melatonin in plants under different physiological states. With the development of isolation and identification technology, numerous evidence shows that plants possess a variety of melatonin-synthesis pathways. The comprehensive application of molecular biology, genomics, and computational biology has also allowed for a comprehensive understanding of the physiological functions of plant melatonin. In spite of the fact that several plant melatonin reviews have been published, the data are still limited and indistinct, and the role of melatonin in plants needs to be further elucidated. In this review, we elucidate and characterize the role of melatonin in plant growth and development in response to biotic and abiotic stresses.

## 2. Biosynthetic Pathways of Melatonin

The biosynthesis of melatonin is essentially divided into two steps, namely tryptophan to serotonin and serotonin to melatonin. The whole process mainly involves six enzymes, namely tryptophan hydroxylase (TPH), tryptophan decarboxylase (TDC), tryptamine 5-hydroxylase (T5H), serotonin N-acetyltransferase (SNAT), N-acetylserotonin methyltransferase (ASMT), and caffeic acid O-methyltransferase (COMT). All of these enzymes, except TPH, have been successfully cloned in plants, in which SNAT was known as Aralkylamine N-Acetyltransferase (AANAT) and ASMT was called hydroxyindole O-methyltransferase (HIOMT) in the previous literature. Scientists used to think that the pathways for melatonin synthesis in animals and plants are the same [2]. However, in subsequent studies, it was gradually discovered that plants have different biosynthetic pathways than animals. Since 2010, there has been a gradual focus on identifying related secretases. The popularity lasted for almost 5 years, and the discovery of an alternative pathway from serotonin to melatonin (serotonin-5-methoxytryptamine-melatonin) became one of the milestones in the research at this stage. Since 2017, scientists have conducted in-depth research on the details, attempting to demonstrate that mitochondria can secrete endogenous melatonin and identifying corresponding synthesis inhibitors [5].

TPH is an enzyme that hydroxylates tryptophan. In 2000, 5-hydroxytryptophan, the precursor of serotonin synthesis, was found in St. John's wort, from which it was concluded that TPH was involved in the synthesis of 5-hydroxytryptophan [6]. Unfortunately, it has not yet been successfully cloned in plants.

TDC belongs to the family of aromatic L-amino acid decarboxylase. This gene was first cloned from periwinkle [7]. Studies have shown that the gene can use tryptophan and 5-hydroxytryptophan as substrates to produce tryptamine and serotonin. Sei Kang et al. cloned TDC into rice and performed corresponding kinetic studies. It was revealed that the protein had high substrate specificity for tryptophan, and the KMichaelis constant (Km) value reached 0.69 mM. Due to the plant's senescence, the production of 5-hydroxytryptophan will be present in the plant. TDC reduces the accumulation rate of 5-hydroxytryptophan, by accelerating the serotonin production of serotonin, resulting in delayed leaves' senescence. Yu Zhao et al. detected the expression of the *PaTDC* gene in sweet cherries through a real-time polymerase chain reaction (RT-PCR) and found that the gene expression was proportionally related to the production of endogenous melatonin in sweet cherries. This finding proved that the TDC gene is involved in the synthesis of plant endogenous melatonin and caused the rate-limiting enzyme in melatonin synthesis [8]. In addition, TDCs may contain a small gene family, with different members exhibiting different activities under different circumstances. Yeong Byeon et al. found that the melatonin concentration in rice seeds overexpressing *TDC3* was 31 times higher than that in wild-type seeds [9]. In contrast, the melatonin content of the transgenic seedlings was only twice that of the wild type. Yeong Byeon et al. speculated that the senescence mechanism of plants enhances the expression of *TDC1* and T5H [10], and inhibition of T5H might indirectly promote the expression of *TDC1* [11].

T5H is a cytochrome P450 enzyme. Schröder et al. have studied the role and synthesis pathway of serotonin. It was found that serotonin is Tryptamine catalizer and converts Tryptamin to serum element [12]. A previous study has successfully cloned and characterized T5H in rice [13]. Furthermore, Sangkyu Par et al. speculated that the T5H pathway could synthesize melatonin through 5-hydroxytryptophan [14]; it should be noted that T5H overexpression does not often lead to the overproduction of serotonin and melatonin, maybe because the catalytic efficiency of T5H is much higher than that of TDC, resulting in a low level of tryptophan in plants under normal circumstances.

In previous studies, scientists have found that the functional activity of SNAT protein is closely related to the synthesis of N-acetylserotonin and melatonin. Endogenous serotonin was successfully activated using Chlamydomonas CrAANAT and human AANAT, by transgenic methods in mini tomatoes and rice, resulting in high melanin content. By analyzing the relative content of metabolites in the process of melatonin synthesis, Sangkyu Park et al.

found that the conversion of serotonin to N-acetyl serotonin was the lowest, and SNAT was assumed to be an important limiting factor in the entire melatonin pathway. Lately, most N-acetylserotonin can be reduced to serotonin under the action of N-acetylserotonin deacetylase (ASDAC) [15], and the involvement of SNAT is higher in the homeostatic regulation of melatonin secretion. Interestingly, SNAT can catalyze the conversion of accumulated tryptophan to N-acetyltryptamine, thereby inhibiting melatonin secretion [14]. Later, N-acetyltryptamine was shown to inhibit the activity of ASMT [16]. Given the large differences in homology of SNAT enzymes in animals and plants, it was found that the family of GCN5-related N-acetyltransferase [17] catalyzes the transfer of acetyl groups from acetyl-CoA to a number of molecules, so it is speculated that plant SNAT plays an important role as a member of the GNAT family. Kiyoon Kang et al. expressed 31 rice GNAT family genes in *E. coli*, studied the activity of SNAT, and finally successfully cloned rice SNAT. Similar studies on enzyme kinetics showed that the maximum reaction rate (Vmax) of SNAT to 5-methoxytryptamine was higher than that of serotonin, leading to the conclusion that serotonin-N-acetylserotonin-melatonin may not be the only way for plants to synthesize melatonin. The earlier study showed that SNAT also intrigued scientists. It was revealed in previous studies that conserved genes resulted in a greater probability to find their ancestors. In contrast to animals, the SNAT of plants is derived from cyanobacteria. Its subcellular localization shows that the enzyme is mainly distributed in chloroplasts [18]. The above results just support the symbiont theory about the possible source of chloroplasts.

ASMT occurs in plants as a small gene family (ASMT1–ASMT3), and the encoded enzymes are mainly found in the cytoplasm [18]. ASMT1 was successfully cloned in rice by Kang et al. [16] in 2011. The study also revealed that this gene has high homology with caffeic acid 3-O-methyltransferase from wheat and is readily induced during aging. Compared with ASMT1, the degree of similarity between ASMT2 and ASMT3 is almost 78%, and the melatonin content of the three related transgenic lines is higher than that of the wild type [19]. Yeong Byeon et al. found in *Arabidopsis thaliana* that *OMT3* gene, which was purified in *E. coli*, has similar functions and activities to ASMT. For this reason, OMT3 is defined as COMT, which is an enzyme and is more inclined to use serotonin as a substrate to generate 5-methoxytryptamine at low temperature (37 °C) [20]. The corresponding enzyme kinetics show that the catalytic efficiency (Vmax/Km) of COMT is 709-fold higher than that of ASMT [21]. Combined with the results of the enzyme kinetics of SNAT, we can conclude that the main synthesis pathway of melatonin in plants is from serotonin to 5-MT, and then 5-MT undergoes acetylation to generate melatonin. This finding was published in 2017 [22].

Current studies have shown that plants can synthesize melatonin in different ways in the mitochondria, chloroplast [23], and cytoplasm [23] (Figure 1). The cytoplasm mainly uses the classical pathway (serotonin-N-acetylserotonin-melatonin) to secrete melatonin. In the chloroplast, melatonin is secreted mainly through an alternative pathway (serotonin-5-methoxytryptamine-melatonin). The successful localization of SNAT in apple mitochondria suggests that mitochondria probably have a similar secretion pathway to chloroplasts. However, a corresponding phylogenetic analysis showed that the MzSNAT5 isolated in apples is more similar to animal SNAT. In conjunction with the findings of Sangkyu Park, it was found that inhibition of T5H expression leads to an increase in 5-hydroxytryptophan (5-OH-Trp) and melatonin levels in plants [11]. It can be concluded that plant mitochondria rely on the (Trp-5-OH-Trp-serotonin-5-methoxytryptamine-melatonin) pathway to secrete melatonin. The accumulation of melatonin biosynthesis intermediates changes under the stimulation of ROS by stress, including cold, drought, salt, cadmium, aging, and pathogens. Since the specificity of the same enzyme for different substrates is different, results of studies on enzyme kinetics and other related studies have shown that the transport of intermediates from the cytoplasm to organelles is limited [22], so we speculated that plants' secretory pathway of melatonin in cells is mainly in the cytoplasm (Figure 1). In addition, depending on the properties of enzymes, there may be other secretion pathways

that may still exist in plants, such as Trp-(TPH)-5-hydroxytryptophan-(methylation)-5-methoxytryptophan-(decarboxylation)-5-methoxytryptamine-(SNAT)–Mel. However, no evidence has yet been found for this metabolic pathway, and the associated content needs further investigation. Finally, we focused further investigation on the key proteins of Arabidopsis secretory pathway and rice in Table 1.

**Table 1.** The key protein in endogenous melatonin synthesis pathway of Arabidopsis and rice.

| No | Protein Name | Gene Name | Organism | Kinetics | Location | Application | References |
|----|-------------|-----------|----------|----------|----------|-------------|------------|
| 1 | Tryptophan decarboxylase 1 | TDC1 | *Oryza sativa* | $K_M$ = 0.69 mM for tryptophan | Cytoplasm | TDC can catalyze the decarboxylation of tryptophan to produce tryptamine and catalyze the decarboxylation of 5-hydroxy-tryptophan to produce serotonin. | [24] |
| 2 | Tryptamine 5-hydroxylase | CYP71P1 | *O. sativa* | - | Endoplasmic reticulum | T5H can catalyze the conversion of tryptamine to serotonin. | |
| 3 | Serotonin N-acetyltransferase 1 | SNAT1 | *O. sativa Panicum virgatum* L. | $K_M$ = 385 µM for serotonin, $K_M$ = 836 µM for tryptamine, $K_M$ = 375 µM for 5-methoxytryptamine | Chloroplast, nucleus | SNAT can catalyze the N-acetylation of serotonin into N-acetylserotonin and catalyze in vitro the N-acetylation of tryptamine to produce N-acetyltryptamine, 5-methoxytryptamine to produce melatonin. | [25] |
| 4 | Serotonin N-acetyltransferase 2 | SNAT2 | *O. sativa* | $K_M$ = 371 µM for serotonin | Chloroplast, cytoplasm | - | [26] |
| 5 | Acetylserotonin O-methyltransferase 1 | ASMT1 | *O. sativa Juglans* | $K_M$ = 864 µM for N-acetylserotonin | Cytoplasm | ASMT can catalyze the transfer of a methyl group onto N-acetylserotonin, producing melatonin. | [18,27] |
| 6 | Acetylserotonin O-methyltransferase 3 | ASMT3 | *O. sativa* | - | Cytoplasm | - | [18] |
| 7 | Acetylserotonin O-methyltransferase 2 | ASMT2 | *O. sativa* | - | Cytoplasm | - | [19] |
| 8 | Flavone 3'-O-methyltransferase 1 | COMT | *O. sativa* | $K_M$ = 243 µM for N-acetylserotonin | Cytoplasm | COMT can catalyze the transfer of a methyl group onto N-acetylserotonin, producing melatonin (N-acetyl-5-methoxytryptamine). | [28] |
| 9 | Flavone 3'-O-methyltransferase 1 | OMT1 | *A. thaliana* | $K_M$ = 233 µM for N-acetylserotonin | Nucleus, cytoplasm, chloroplast, plasmodesma | - | [21] |

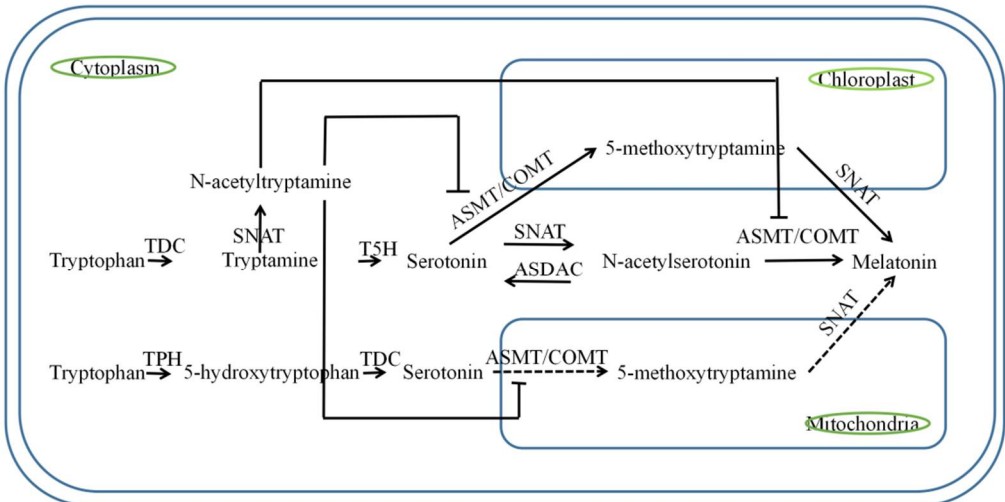

**Figure 1.** Biosynthetic pathway of melatonin in plants.

## 3. Role of Melatonin in Plants

### 3.1. Regulatory Role of Melatonin in Plants

#### 3.1.1. Improve Seed Germination Rate

Seed germination occurs by red light (PR-inactive form; 666 nm) and far-red light (PFR-active form; 730 nm). As the structure of the dimer proteins changes, the seeds convert light signals into biological signals that regulate germination. Due to their amphiphilic and antioxidant properties, melatonin and its derivatives can easily penetrate the seed throughout the germination process to protect the lipids of the cell membrane from peroxidation, thus improving seed vigor and germination rate [29]. Moreover, the process of seed germination will be accompanied by a large amount of sugar metabolism, and the addition of melatonin alters the gluconeogenesis pathway of cells to promote the conversion of amino acids into starch [30]. Tomato (*Solanum lycopersicum* L.) seeds were found to accumulate a large amount of melatonin before germination [31]. The addition of exogenous melatonin during the treatment of cabbage seeds with water increased the percentage of seed germination (FGP) from 71.7% to 87.5% [32]. Under low-temperature conditions, the combination of osmotic priming and melatonin treatment of cucumber seeds can increase the seed germination rate from 4% to 98% [29]. In the dark at 30 °C, the induction of seeds of *Phacelia tanacetifolia* with 6 μM exogenous melatonin increased the FGP from 2.5% to 52% [33]. Treatment of sterile cucumber (*Cucumis sativus* L.) seeds with different concentrations of melatonin solution revealed that melatonin could effectively reduce the inhibitory effect of polyethylene glycol [34] on seed germination; a similar result was also observed in cotton (*Gossypium hirsutum* L.) [35] and Arabidopsis [36]. These results showed that 20 μM exogenous melatonin is the optimal dose to promote cotton seed germination. Moreover, melatonin can integrate abscisic acid, gibberellin, and auxin in seed germination. Recently, Yu et al. [37] also found that 100 μM melatonin can improve the heat resistance of rice seeds and increase their germination rate at high temperature.

#### 3.1.2. Rooting Promotion

Plant roots include seed roots, adventitious roots, and lateral roots. Seed roots develop during embryogenesis, and adventitious roots and lateral roots develop after embryogenesis. Current research shows that most of the characteristics of root development are regulated by auxin. Plant melatonin has the same synthetic precursor serotonin as indoleacetic acid (IAA). Thus, low melatonin concentrations can promote adventitious and lateral root formation, while high melatonin concentrations have an inhibitory effect. During root development of rice seedlings, internal melatonin is decomposed into 2-hydroxymelatonin (2-OHM) and 3-hydroxymelatonin (3-OHM) under the action of melatonin 3-hydroxylase (M2H). This could be the main reason for the relatively low melatonin content in plant roots.

Studies on lupine tissue have shown that the two endogenous hormones have similar concentration profiles. In studying the effect of melatonin on the rooting of commercial cherry rootstocks, it was found that 1 μM melatonin can promote root number and length, while 10 μM melatonin has an inhibitory effect on the rooting of the rootstock vines tested [38]. Ramo'n Pelagio-Flores [39] analyzed the expression of auxin-responsive genes in transgenic Arabidopsis seedlings and found that auxin-responsive promoters (DR5:uidA and BA3:uidA) and HS, heat shock promoter (HS::AXR3NT)-GUS), expression did not change significantly, so it was inferred that exogenous melatonin would act independently of auxin during root development. Furthermore, it was found that the enhancement of seminal root and root biomass in transgenic rice could not affect the adventitious root development. Therefore, Sangkyu Park assumed that melatonin is unlikely to be involved in the process of rice auxin synthesis or auxin signal transduction [40]. The research results in tomato showed that 100 μM melatonin could alleviate the environmental stress on the roots of seedlings under drought stress. This improved the survival rate of tomato [41]. In the study of apple rootstocks, Du et al. [42] also found that exogenous melatonin could alleviate low nitrate stress by changing root structure, promoting nitrate uptake, and regulating the expression of genes related to nitrogen transport and metabolism.

### 3.1.3. Regulation of Plant Growth and Development

In addition to the non-enzymatic degradation of melatonin by common oxidants, there are also some enzymatic degradation pathways. Studies have found that indoleamine 2,3-dioxygenase (IDO) in water hyacinth can metabolize melatonin into (N1-acetyl-N2-formyl-5-methoxykynuramine) AFMK from time to time [43]. Using the rice *OsIDO* gene, constructed transgenic tomato plants not only had lower melatonin content but also had a reduced number of lateral leaves. This fully indicates that the interaction between IDO and melatonin will affect the development of plant leaves [44]. Flowering is an important sign of plants from the vegetative stage to the reproductive stage, and GA is an essential substance in the flowering process. In the presence of GA, GA will bind to the receptor GID1. The function of GID1 is to mediate the degradation of DELLA protein. Melatonin can ensure the stability of DELLA protein as much as possible, without affecting the level of endogenous GA, thereby delaying the flowering period [45]. Under the same planting conditions, transgenic rice constructed by overexpressing sheep serotonin N-acetyltransferase showed enhanced seedling growth, delayed flowering, and reduced spikelets per panicle, compared with the wild type [46]. In addition, melatonin also exhibits concentration characteristics in the process of plant growth and development. Zhao et al. found that applying low concentration of melatonin to maize could enhance photosynthesis and sugar metabolism in the phloem of maize seedlings [47]. High concentration of exogenous melatonin could inhibit the growth of maize seedlings. Furthermore, melatonin can induce bud formation in the same manner as the study of Erland et al. [48]. The involvement of melatonin in the dark growth of plants has also been an important piece of research in recent years. It has been found that inhibiting SNAT2, which is often expressed at night in rice, will lead to the dwarfing of plants. A similar phenomenon has also been found in apples. Melanin can be mediated by light signals to regulate the flowering period of apples in a dose-dependent manner [49], and endogenous MT levels in peppers also increase under light conditions [50]. Strigolactone (SL) appears to have an inhibitory effect on melatonin secretion, when impaired SL synthesis signaling and transduction in Arabidopsis mutants d14-1 and max4-1 results in multiple genes encoding MT synthase. Thus, the expression level is up-regulated. Li et al. [51] also found that MT can induce the oxidative stress induced by environmental copper ions, and reintegrate various defenses and systems within rice to ensure normal growth of rice seedlings. Finally, it should also be noted that some researchers believe that melatonin can promote plant growth by regulating carbon assimilation and ATP accumulation [52].

### 3.1.4. Reproductive Regulation of Plants

Studies have shown that melatonin has a protective effect on the reproductive tissue of plants, and its content changes with the maturity of the fruit and flower tissue. Rice accumulates the melatonin synthesis precursor tryptophan in flowers at the production stage, and the melatonin content in flower spikes is six times higher than that in the related flag leaves [53]. The mother plant of wheat secretes melatonin at the grain-filling stage, which can increase the activity of antioxidant enzymes in the progeny and improve the cold tolerance of seedlings. In an early study, it was found that the melatonin content was highest in the early stage of flower-bud development, and it gradually decreased as the flower buds matured in datura flowers. The content remained stable at a high level [54]. The melatonin content in the skin of grapes increased to 45% of fully developed purple before breeding [55]. In contrast, melatonin in the skin, seeds, and pulp is relatively reduced during the breeding process. [56]. Melatonin-treated lychees had an increase in total phenolic, flavonoid, and anthocyanin content, thereby delaying their own postharvest browning. The concentration of MT was higher in bell pepper seedlings but then gradually decreased [57], while the MT content was highest at the tip of mulberry leaves, and the young leaves had higher content than the old leaves [58]. Moreover, during the ripening process of the "Red Fuji" apple (Malus domestica Borkh. cv. Red), an increase in fruit respiration intensity leads to an increase in reactive oxygen species, and the trend of their change parallels the trend of change in the associated endogenous melatonin content [59]. Foreign melatonin can effectively induce the cell-wall-modifying proteins polygalacturonase [60], pectin esterase1 (PE1), β-galactosidase (TBG4), and expansin1 (Exp1); fruit color development genes, such as phytoene synthase1 (PSY1) and carotenoid isomerase (CRTISO); the expression of aquaporin genes such as SlPIP12Q, SlPIPQ, SlPIP21Q, and SlPIP22; enhanced ethylene production; and fruit softening and ripening.

### 3.1.5. Alleviating Effect on Plant Senescence

Plant senescence is an intricate mechanism. Hormones, such as jasmonic acid (JA), ethylene (ET), abscisic acid (ABA), and salicylic acid (SA), and external factors, including photoperiod, biotic stress, abiotic stress, and nutrient limitation, activate the expression of senescence-associated genes (SAGs). With the decrease in the stability of chlorophyll–protein complexes, a large amount of chlorophyll will be released and degraded under the action of CLH (chlorophyllase, a chlorophyll hydrolase) and PAO (pheophorbide aoxygenase, a ferredoxin-dependent monooxygenase). Current research suggests that senescence is caused by an uncontrolled surge of ROS, and accumulation of hexose sugars is a signal for the onset or acceleration of senescence. An Arabidopsis model shows that the hexokinase gene (*AtHXK1*) acts as a sugar sensing and induces signaling in senescence mechanism. In the initial stage of plant senescence, PAO, CLH, and chlorophyll will act synergistically to reduce the accumulation of ROS and other phototoxic molecules [61]. This function, thus, parallels the antioxidant effect of melatonin at this level of reducing reactive oxygen species accumulation. Melatonin will maintain the ascorbic acid-glutathione (AsA-GSH) cycle by controlling AUXIN RESISTANT 3 (AXR3)/INDOLE-3-ACETIC ACID INDUCIBLE 17 (IAA17) isogenic ABF, ABI, and other transcription factors that indirectly control the expression of senescence genes, such as HXK1 and autophagy-related genes, thereby slowing down the rate of chlorophyll degradation [62]. At the same time, studies in kiwifruit also found that melatonin can also delay the senescence of kiwifruit leaves through the transcription level of the genes involved in flavonoid synthesis [63]. In the process of fruit preservation at low temperature, MT was also found to slow down the senescence rate of the fruit after repicking, by increasing the activity of NO synthase and inhibiting the respiration and ethylene production rate of the fruit. Cucumber findings suggest [64] that MT will act as a powerful antioxidant to alleviate leaf senescence by activating the antioxidant system and IAA synthesis and signaling, while inhibiting ABA synthesis and signaling in cucumber plants.

### 3.2. Protective Role of Melatonin in Plants

Free radicals are a class of substances with unpaired electrons that can rapidly react with almost any molecule in a living cell. Low concentrations of free radicals can not only induce mitosis but also induce a necessary substance in the maturation of cell structures. However, once its concentration upsets the balance between production and consumption, it can lead to oxidative stress in plants [65]. Under the stimulation of oxidative stress, the efficiency of the plant's own Calvin cycle decreases, and the rate of NADPH consumption will gradually slow down. Ferredoxin produced during photosynthetic electron transport will accelerate the production of reactive oxygen species (ROS) in the chloroplast. In addition, ROS can also be generated in the event of leakage of electron transport within mitochondria.

Plants will scavenge excess free radicals in cells by increasing the activity of relevant antioxidant enzymes or the concentration of non-enzymatic antioxidants. Enzymatic and non-enzymatic antioxidant like catalase (CAT), superoxide dismutase (SOD), glutathione peroxidase (GSH-Px), carotenoids, and ascorbic acid can metabolize these ROS. Studies have shown that melatonin can increase the transcription efficiency of these antioxidant enzymes' mRNAs by activating transcription factors in the promoter regions of antioxidant enzymes gene, resulting in the enhancement of the antioxidant level. In addition, the cascade reaction of melatonin and its metabolites also provide free-radical scavenging effects which protect plants at low melatonin concentration. Therefore, it can be said that melatonin is the endogenous free radical scavenger with the strongest antioxidant effect currently known.

Autophagy is another self-defense process of plant cells against oxidative stress. Organelles that suffer from oxidative damage will be delivered to vacuoles or lysosomes. Current research shows that plant cells include microautophagy (the cytoplasm is directly phagocytosed by the vacuolar membrane) and macroautophagy (the cytoplasm is surrounded by double-membrane structures to form autophagosomes). Studies in Arabidopsis have shown that melatonin is likely to participate in the plant's own AtATG8-PE conjugation pathway, thereby enhancing its own autophagy [66]. In addition, the latest research results show that melatonin can induce related apoptosis by inhibiting the inhibition of the p38 MAPK signaling pathway [67]. Adversity such as drought, low temperature, heavy metals, UV radiation, and biotic stress can cause oxidative damage to plant cells, and melatonin plays an important role in countering these damages. Sometimes, the metabolites produced by melatonin can even alter the flora of the soil near the plant, with some unexpected effects [68].

### 3.2.1. Improve Plant Resistance to Cold Damage

Melatonin has the ability to increase the resistance of plants to cold damage. The MT content in field tomato leaves was about 10 times higher than that in greenhouse seedlings [69]. Early studies have shown that exogenous melatonin-treated carrot suspension cells induced by low temperature can alleviate the apoptosis caused by cold injury, by increasing the level of intracellular polyamines including putrescine and spermidine [70]. Cryopreservation of *Rhodiola grandiflorum* callus pretreated in medium containing 0.1 μM melatonin resulted in a significantly higher survival rate than untreated tissue [71].

In the ABA pathway, low temperature signals were transmitted by plants resulting in the induction of CBF expression (ICE)—C-repeat binding factor (CBF)—in the cold-regulated genes (COR) pathway. The COR gene of high-freezing-resistant plants tended to be higher expressed than non-freezing-tolerant plants. Previous study confirmed that exogenous melatonin and low-temperature treatment induces a significant increase in the expression of endogenous melatonin as well as *EnCBFs*, *EnCOR14a*, and ABA level. It clearly showed that melatonin will cross-interact with ABA synthesis signals to improve the cold/freezing resistance in plants. In the Arabidopsis model, we found that melatonin, in addition to activating the expression of CBFs and COR15a, induces transcriptional activators such as CAMTA1 (a transcription factor involved in freezing- and

drought-stress tolerance), ZAT10, and ZAT12 (antioxidant gene-transcription activator) expression [72]. Shi et al. also found that the CBF pathway activated by ZAT6 is involved in melatonin-mediated freezing stress in Arabidopsis. In view of the broad application prospects of melatonin in improving plant cold tolerance, some research teams have also used melatonin to reduce the stimulation of rare species preserved in ultra-low-temperature environments [73].

### 3.2.2. Improve Plant Resistance to Drought Stress

Under drought stress, plants reduce water loss by secreting abscisic acid (ABA), which, in turn, induces their own cells to retain water and nutrients during the reproductive cycle. This inhibits the accumulation of nutrients in the leaves, thereby accelerating leaf senescence and shedding. Ping Wang [74] added 100 μM melatonin in arid soil, which can effectively inhibit the expression of SAG12 and PAO and delay the leaf senescence caused by drought. The lack of water also often affects stomatal size, resulting in a large accumulation of ROS in leaves. Melatonin can not only act on the receptors on stomatal (CAND2/PMTR1), but also can effectively neutralize stomatal size. Drought stress was simulated by applying 10% PEG to grape cuttings. Applying melatonin to cuttings not only maintains the stability of photosynthetic electron transport in photosystem II (relieving the effects of oxidative stress to a certain extent) but also mitigates the effects of PEG on cell ultrastructure [75]. When drought is accompanied by high temperature, the expression of the plant's own class A1 heat shock factors (HSFA1s) is often induced. Melatonin can largely inhibit this gene and some common heat-responsive genes' (HSFA2, HSA32, HSP90, HSP101) expression and post-expression ubiquitination. Li et al. [76] proved that *OsSGT1* and *OsABI5* are the keys for MT to improve drought stress. Once seedlings were mutated in *OsSGT1* and *OsABI5*, the MT-mitigation effect was greatly diminished. Wang et al. found [77] that if melatonin and $H_2S$ are mixed, they can better cope with osmotic stress caused by drought. Exogenous melatonin promotes the expression of genes related to nitrogen metabolism in cotton under drought stress [78].

### 3.2.3. Improve the Anti-Insect and Antibacterial Ability of Plants

When plant cells are infected by bacteria, the balance of reactive oxygen species and NO in the cells will be broken. Under the dual stimulation of NO and ROS, plant cells enhance the secretion of their own endogenous melatonin by activating the MAPK cascade signaling pathway (MAPKK4, MAPKK5, MAPKK7, and MAPKK9 activate MAPK3 and MAPK6) (activation of the RAV gene family). It may also be the key to plant melatonin secretion [79]. Melatonin can regulate bacterial proliferation/replication. The rapid growth of bacteria is highly dependent on the free extracellular iron ions. Melatonin, on the other hand, has a high metal-binding capacity, and when it acts on the cell wall, it will effectively limit the vitality of bacteria. In addition, when melatonin acts on the cell membrane, it also prevents bacteria from taking up certain stimulators (often fatty acids) from the outside world, reducing the activation of the genes that promote cell proliferation [80]. Melatonin can also induce HSP90s [74] and related transcription factors, and activate defense genes including plant defensin 1.2 (PDF1.2), isochorismate synthase 1 (ICS1) under the synergistic action of a series of hormones IAA, SA, and ET, and ascorbate peroxidase 1 (APX1)) to defend against damage caused by pathogens such as *Pseudomonas syringae* DC3000, *Podosphaera xanthii*, *Colletotrichum musae*, and *Phytophthora capsici* [81]. Based on previous findings, the use of exogenous melatonin effectively helps banana, cassava, and other plants to relieve the limitation of physiological deterioration (PPD). Plant cells also defend against pathogens by enhancing the thickness of cell walls, while melatonin inhibits the activity of specific invertase inhibitor proteins (C/VIF) and promotes the expression of cell wall invertase (CWI). Apple replant disease (ARD) tends to lead to a decrease in apple yield, and the application of melatonin also helps to alter the composition of bacterial and fungal communities in the soil, thereby mitigating the effects of ARD [82].

### 3.2.4. Improving Plant Resistance under Chemical Contamination

Melatonin functions to protect plants against chemicals damage and heavy metal. ROS production is generated by peroxidative herbicides which causes plant died. On the other hand, melatonin-rich transgenic (MRT) rice plants exhibit high resistance to peroxidative herbicides. Arnao et al. [83] treated barley plants with sodium chloride, zinc sulfate, and hydrogen peroxide, and found that the melatonin content in the roots of the stressed plants was 6 times higher than that in the roots of the control group. Metal ions such as cadmium, lead, copper, and aluminum are easily reduced by other substances in the living body, such as ascorbic acid, to cause the Fenton reaction. The whole process will produce a large amount of OH. Melatonin not only effectively chelates these metal ions: its metabolites, AFMK, AMK and 3OHM, can also effectively inhibit the production of OH. Li et al. [84] focused on the expression of ion channel genes and found that melatonin would increase the expression of antiporters MdNHX1 and MdAKT1 on the vacuoles of *Malus hupehensis* Rehd, maintain ion homeostasis in leaves, and alleviate ion imbalance for plants. At the same time, melatonin also increases the expression of the high-affinity K transporter (HAK) and CBL1-CIPK23 pathway genes, and promotes the absorption of potassium ions in plants [85]. Excessive use of nitrogen fertilizers often leads to an imbalance of elements in fruits and vegetables. In this case, appropriate pretreatment of seedlings with melatonin can effectively inhibit the accumulation of nitrate and increase the activity of nitrate metabolizing enzymes [86]. Li et al. found that combination co-treatment of selenium (Se) and melatonin under cadmium (Cd) stress significantly reduces the detrimental effects of Cd stress. Farooq et al. [87] used a new kind of novel complex MT-Se nanoparticles (MT-Se NPs) to mitigate the effects of arsenic on canola. Zhang et al. [88] found that MT can enhance the salt tolerance of wheat seeds and seedlings by regulating the synthesis of soluble proteins and sugars, which regulates the ion balance in the body and enhances the antioxidant enzyme system.

### 3.2.5. Photoprotection

Ultraviolet light (UV) is a potent inducer of ROS. The singlet oxygen ($^1O_2$) induced by UV. $^1O_2$ degrade the cell membrane and as oxidize DNA. Due to its lipophilic characteristics, melatonin can easily pass through biological membranes and reduce cell damage caused by $^1O_2$ [89]. As an organic compound, melatonin also dissociates from its own covalent bond and undergoes photolysis after absorbing light wave energy [90]. However, an early study found that compared with melatonin, its main photolysis product AFMK (N1-acetyl-N2-formyl-5-methoxykynuramine) has stronger hydroxyl radical (*OH) scavenging ability. Further, it was confirmed that AFMK hydrolyzed to AMK (N(1)-acetyl-5-methoxykynurenin), another kind of potent singlet oxygen scavenger. It was assumed melatonin has the ability to resist UV damage in higher plants. Moreover, melatonin concentration in plant root tissues was significantly increased under high-intensity UV-B radiation, compared to low-intensity UV-B radiation stimulation in *Glycyrrhiza uralensis* [91]. When observing the fluctuation law of melatonin content in grapes in the day/night cycle, the melatonin content of grapes is affected by sunlight, since during the day it will decrease compared with the night. The resulting malondialdehyde content also showed the same trend [92]. Recently, Bychkov et al. discovered [93] a melatonin-mediated photooxidative receptor CAND2/PMTR1 in *Arabidopsis thaliana*.

Interestingly, Maharaj [94] found that under high-energy laser irradiation, melatonin was oxidized to generate $^1O_2$. The previous study also found that the pH in plants also determines the speed of HRP (plant enzyme horseradish peroxidase), catalyzing the oxidation of melatonin [95]. This evidence fully demonstrates that although melatonin can be used as an efficient free radical scavenger in plants, it is also a free radical generator.

## 4. Conclusions and Future Perspectives

Melatonin is a small molecule compound in plant cells. It acts as a signal molecule, growth regulator, and antioxidant in plants. Plants mainly rely on the cytoplasm to secrete

melatonin, and the whole secretion pathway is complex and diverse. This paper summarizes and sorts out the latest progress of melatonin research in recent years, introducing the basic information on melatonin, the secretion pathway, and the interaction between melatonin and other hormones under different physiological functions. Unfortunately, compared with other hormones in plants, melatonin has many aspects to be studied:

(1) At present, in addition to rice and Arabidopsis, scientists still lack information on the biosynthetic genes and related subcellular localization in other plants;
(2) Plants have a variety of melatonin biosynthesis pathways, but the key regulators in these secretion pathways remain to be determined;
(3) The exact position of melatonin in the whole hormonal interaction network remains to be further studied.

The above questions are still inconclusive, so further research on them will help us understand more about plant melatonin.

**Author Contributions:** R.S. contributed to writing and original draft preparation; F.N.R. and H.Y. edited the manuscript; C.D. and X.Z. contributed to supervision, project administration, funding acquisition, review, and editing the manuscript. All authors have read and agreed to the published version of the manuscript.

**Funding:** This review was funded by the National Key Research and Development Program of China grant number 2021YFD2201204.

**Institutional Review Board Statement:** Not applicable.

**Acknowledgments:** The authors appreciate the reviewers for their comments and suggestions.

**Conflicts of Interest:** The authors report no conflicts of interest.

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
