# Peer review of "Plant Melatonin: Regulatory and Protective Role"

_horticulturae, doi:10.3390/horticulturae8090810_

Round 1
Reviewer 1 Report
Dear Authors
At present, the understanding of biosynthetic pathway of endogenous melatonin in plants has been improved but still need to be explored through a physiological model such as the model of regulation plant growth and development, stress signaling pathways, and interactions with plant hormones. Although several review about plant melatonin have been emphasized, the data is still limited and indistinct, and the role of melatonin in plants needs to be further elucidated. In an article submitted for review entitled ,,The plant melatonin: regulatory and protective role’’, the role of melatonin in plant growth and development in response to biotic and abiotic stresses was characterized and explained. Considering the above, the topic of this review article is well chosen.
However, I have some comments that should be taken into account by the Authors when revising this manuscript.
1. The most important flaw in the manuscript is the difficulty or even inability to identify the author of the presented content (Lines: 109-115; 137-139; 154; 193; 280-290; 298-302; 325-328; 328-329; 370-373; 411-416; 439-447; 451; 495-500; 505-207; 523.
2. References are usually well chosen, except for a very outdated item [19] Balemans, M.G .; Ebels, I .; Hendriks, H.G .; van Berlo, M .; de Moreé, A. The influence of some pteridines on pineal 5-methoxyindole synthesis in male Wistar rats periodically exposed to either white or green light. Journal of neural transmission 1983, 58, 121-134, doi: 10.1007 / bf01249130 ''
3. In general, the statements and conclusions made by the authors in Abstract are consistent and supported by listed citations. However, I have reservations about the content of the Conclusion chapter. The authors focused on pointing to the gaps in the current knowledge, but did not sufficiently refer to the substantive content presented in their manuscript.
4. Please complete the Reference for no 1 in Table 1.
Author Response
Response Letter
Reviewer 1 of manuscript horticulturae-1865841
Horticulturae Journal
August 16th, 2022
Dear Reviewer 1,
Thank you for your comments and the opportunity to revise our manuscript on
Horticulturae Journal. The comments offered by reviewer 1 has been immensely helpful, and we also appreciate all insightful comments on revising all aspects of the previous manuscript.
I have included the reviewer 1 comments immediately below this letter, indicating exactly how we addressed each concern or problem and describing the changes we have made. The revisions have been approved by all authors.
We hope the revised manuscript will better suit to reviewer 1 and Horticulturae Journal but we are happy to consider further revisions. We thank you for your acceptance and interest in our research.
Sincerely,
Corresponding author
We hope the revised version of our manuscript will be better than previous manuscript.
- Comments :The most important flaw in the manuscript is the difficulty or even inability to identify the author of the presented content (Lines: 109-115; 137-139; 154; 193; 280-290; 298-302; 325-328; 328-329; 370-373; 411-416; 439-447; 451; 495-500; 505-207; 523.
Response: Thank you for your comments. I'm sorry for the inconvenience. We have made a comprehensive adjustment to the article from three aspects of English words, sentences and grammar. The full text is rewritten in a simpler and calmer style
Changes : Full text rewriting
- Comments :References are usually well chosen, except for a very outdated item [19] Balemans, M.G .; Ebels, I .; Hendriks, H.G .; van Berlo, M .; de Moreé, A. The influence of some pteridines on pineal 5-methoxyindole synthesis in male Wistar rats periodically exposed to either white or green light. Journal of neural transmission 1983, 58, 121-134, doi: 10.1007 / bf01249130 ''
Response: Thanks for the comments. We have referred to your suggestions, revised the article and conducted a new review of all references.
Changes : References
3.Comments : In general, the statements and conclusions made by the authors in Abstract are consistent and supported by listed citations. However, I have reservations about the content of the Conclusion chapter. The authors focused on pointing to the gaps in the current knowledge, but did not sufficiently refer to the substantive content presented in their manuscript.
Response: Thanks for the comments. We have revised the Conclusions and Future Perspectives of the article.
4.Comments : Please complete the Reference for no 1 in Table 1.
Response: Thanks for the comments. We have referred to your suggestions, revised the article and conducted a new review of all references.
Changes : Table 1

Reviewer 2 Report
please see the attachment

Author Response
Response Letter
Reviewer 1 of manuscript horticulturae-1865841
Horticulturae Journal
August 16th, 2022
Dear Reviewer 2,
Thank you for your comments and the opportunity to revise our manuscript on
Horticulturae Journal. The comments offered by reviewer 1 has been immensely helpful, and we also appreciate all insightful comments on revising all aspects of the previous manuscript.
I have included the reviewer 1 comments immediately below this letter, indicating exactly how we addressed each concern or problem and describing the changes we have made. The revisions have been approved by all authors.
We hope the revised manuscript will better suit to reviewer 2 and Horticulturae Journal but we are happy to consider further revisions. We thank you for your acceptance and interest in our research.
Sincerely,
Corresponding author
We hope the revised version of our manuscript will be better than previous manuscript.
- Comments :please see the attachment
Response: Thank you for your comments. Sorry for the inconvenience. We have made a comprehensive adjustment to the article in terms of English words, sentences and grammar with reference to your suggestions. The full text is rewritten in a simpler and calmer style
Changes : Full text rewriting

Reviewer 3 Report
The authors propose a review on the biosynthetic pathways and the role of melatonin on the development of plants and their response to stress. The idea of ​​a review on this subject is original and deserves to be published in a review like Horticulturae. However, the article looks like a draft version and needs to be completely rewritten before being resubmitted. Both English syntax, punctuation and typing errors need to be corrected. Some sentences have no verb, other expressions are repeated twice, the typology of the text is not correct, the presentation of some data is not referenced, and many results are overinterpreted. It seems that the version of the article submitted corresponds to a first draft. I consider that although the subject is original and that a review on the implication of melatonin in plants has its place to be published in Horticulturae, it is essential before a possible resubmission that the article be completely rewritten in a understandable English and in a more sober style. A summary diagram of the role of melatonin on the development of the plant and its response to stress would be welcome.
Author Response
Response Letter
Reviewer 3 of manuscript horticulturae-1865841
Horticulturae Journal
August 16th, 2022
Dear Reviewer 3,
Thank you for your comments and the opportunity to revise our manuscript on
Horticulturae Journal. The comments offered by reviewer 1 has been immensely helpful, and we also appreciate all insightful comments on revising all aspects of the previous manuscript.
I have included the reviewer 1 comments immediately below this letter, indicating exactly how we addressed each concern or problem and describing the changes we have made. The revisions have been approved by all authors.
We hope the revised manuscript will better suit to reviewer 3 and Horticulturae Journal but we are happy to consider further revisions. We thank you for your acceptance and interest in our research.
Sincerely,
Corresponding author
We hope the revised version of our manuscript will be better than previous manuscript.
- Comments :The authors propose a review on the biosynthetic pathways and the role of melatonin on the development of plants and their response to stress. The idea of a review on this subject is original and deserves to be published in a review like Horticulturae. However, the article looks like a draft version and needs to be completely rewritten before being resubmitted. Both English syntax, punctuation and typing errors need to be corrected. Some sentences have no verb, other expressions are repeated twice, the typology of the text is not correct, the presentation of some data is not referenced, and many results are overinterpreted. It seems that the version of the article submitted corresponds to a first draft. I consider that although the subject is original and that a review on the implication of melatonin in plants has its place to be published in Horticulturae, it is essential before a possible resubmission that the article be completely rewritten in a understandable English and in a more sober style. A summary diagram of the role of melatonin on the development of the plant and its response to stress would be welcome.
Response: Thank you for your comments. Sorry for the inconvenience. We have made a comprehensive adjustment to the article in terms of English words, sentences and grammar with reference to your suggestions. The full text is rewritten in a simpler and calmer style
Changes : Full text rewriting

Round 2
Reviewer 2 Report
the revised manuscript is ready for the publication.